# Risk factors for overtaking, rear-end, and door crashes involving bicycles in the United Kingdom: Revisited and reanalysed

Chun-Chieh Chao[1,2,3], Hon-Ping Ma[1,3,4], Li Wei[1,5,6], Yen-Nung Lin[1,7], Chenyi Chen[1], Wafaa Saleh[8], Bayu Satria Wiratama[9], Akhmad Fajri Widodo[1], Shou-Chien Hsu[10,11], Shih Yu Ko[4], Hui-An Lin[1,2,3], Cheng-Wei Chan[1,10,12,13], Chih-Wei Pai[1] *

1 Graduate Institute of Injury Prevention and Control, College of Public Health, Taipei Medical University, Taipei City, Taiwan, 2 Department of Emergency Medicine, Taipei Medical University Hospital, Taipei City, Taiwan, 3 Department of Emergency Medicine, School of Medicine, College of Medicine, Taipei Medical University, Taipei, Taiwan, 4 Department of Emergency Medicine, Taipei Medical University-Shuang Ho Hospital, New Taipei City, Taiwan, 5 Taipei Neuroscience Institute, Taipei Medical University, Taipei, Taiwan, 6 Department of Surgery, Division of Neurosurgery, Wan Fang Hospital, Taipei Medical University, Taipei, Taiwan, 7 Department of Physical Medicine and Rehabilitation, Wan Fang Hospital, Taipei Medical University, Taipei, Taiwan, 8 Transport Research Institute, Edinburgh Napier University, Edinburgh, Scotland, 9 Faculty of Medicine, Department of Epidemiology, Biostatistics and Population Health, Public Health and Nursing, Universitas Gadjah Mada, Yogyakarta City, Indonesia, 10 Department of Emergency Medicine, Chang Gung Memorial Hospital, Linkou Branch, Taoyuan, Taiwan, 11 Department of Occupational Medicine, Chang Gung Memorial Hospital, Linkou Branch, Taoyuan, Taiwan, 12 Department of Emergency Medicine, New Taipei City Hospital, New Taipei City, Taiwan, 13 College of Medicine, Chang Gung University, Taoyuan City, Taiwan

☯ These authors contributed equally to this work.
* cpai@tmu.edu.tw

**Data Availability Statement:** This study utilised the British STATS19 database, which contains data on all road traffic accidents in the United Kingdom. The data that support the findings of this study are

## Abstract

### Background and objective

Relevant research has provided valuable insights into risk factors for bicycle crashes at intersections. However, few studies have focused explicitly on three common types of bicycle crashes on road segments: overtaking, rear-end, and door crashes. This study aims to identify risk factors for overtaking, rear-end, and door crashes that occur on road segments.

### Material and methods

We analysed British STATS19 accident records from 1991 to 2020. Using multivariate logistic regression models, we estimated adjusted odds ratios (AORs) with 95% confidence intervals (CIs) for multiple risk factors. The analysis included 127,637 bicycle crashes, categorised into 18,350 overtaking, 44,962 rear-end, 6,363 door, and 57,962 other crashes.

### Results

Significant risk factors for overtaking crashes included heavy goods vehicles (HGVs) as crash partners (AOR = 1.30, 95% CI 1.27–1.33), and elderly crash partners (AOR = 2.01, 95% CI = 1.94–2.09), and decreased risk in rural area with speed limits of 20–30 miles per hour (AOR = 0.45, 95% CI = 0.43–0.47). For rear-end crashes, noteworthy risk factors

openly available at https://figshare.com/ndownloader/files/48173452.

**Funding:** This study was financially supported by grants from the Ministry of Science and Technology, Taiwan (MOST 109–2314-B-038-066-); the National Science and Technology Council, Taiwan (NSTC 112-2410-H-038-023-MY2; NSTC 110-2410-H-038-016-MY2) and New Taipei City Hospital (NTPC 113-002). The funders had no role in the design of the study, data collection and analysis, interpretation of data, or preparation of the manuscript.

**Competing interests:** The authors have declared that no competing interests exist.

**Abbreviations:** WHO, World Health Organization; HGVs, heavy goods vehicles; AOR, adjusted odds ratio; CI, confidence interval.

included unlit darkness (AOR = 1.49, 95% CI = 1.40–1.57) and midnight hours (AOR = 1.28, 95% CI = 1.21–1.40). Factors associated with door crashes included urban areas (AOR = 16.2, 95% CI = 13.5–19.4) and taxi or private hire cars (AOR = 1.61, 95% CI = 1.57–1.69). Our joint-effect analysis revealed additional interesting results; for example, there were elevated risks for overtaking crashes in rural areas with elderly drivers as crash partners (AOR = 2.93, 95% CI = 2.79–3.08) and with HGVs as crash partners (AOR = 2.62, 95% CI = 2.46–2.78).

## Conclusions

The aforementioned risk factors remained largely unchanged since 2011, when we conducted our previous study. However, the present study concluded that the detrimental effects of certain variables became more pronounced in certain situations. For example, cyclists in rural settings exhibited an elevated risk of overtaking crashes involving HGVs as crash partners.

## Introduction

In recent years, urban bicycling has become increasingly popular in many countries, offering benefits such as reduced traffic congestion, diminished parking pressure, and a reduction in greenhouse gas emissions [1,2]. The World Health Organization has highlighted numerous health advantages of moderate-intensity physical activities such as bicycling, including improvements in life expectancy, quality of life, cognitive function, mental health, sleep quality, muscular and cardiorespiratory fitness, and bone and functional health [1].

However, despite such health benefits, the risk of injury remains a considerable safety concern for cyclists, who are regarded as vulnerable road users [1,3]. Traffic crash data indicate that the risk of accidents for cyclists, measured per distance travelled, is approximately 20 times higher than that for vehicle drivers [1]. To address this problem, researchers in the United States developed a comprehensive bicycle route safety rating model with a focus on injury severity [4]. This model evaluates multiple operational and physical aspects such as traffic volume, population density, highway classification, lane width, and the presence of one-way streets. In addition, it is capable of predicting the severity of injuries due to motor vehicle–related crashes at specific locations [4]. Another finding was that a route is considered adequately safe if it includes geometric factors that enhance safety [4]. This model can aid urban planners and public officials in creating infrastructure such as bike lanes and implementing strict lane policies to improve cyclist safety [4]. Implementing bike lanes has been demonstrated to reduce crash rates by up to 40% among adult cyclists [5]. One study found that roundabouts with dedicated cycle tracks significantly lower the risk of injury for cyclists compared to those without such bicycle infrastructure [6]. Furthermore, adequate night-time lighting on rural roads has the potential to prevent over half of all cyclist injuries [7]. Bicycle crashes can also impose a significant burden on healthcare expenses. Elvik and Sundfør [8] have discussed the economic implications and healthcare expenditures associated with bicycle accidents. For instance, in Belgium, the average cost of bicycle accidents per case is estimated at 841 euros [9]. In the Netherlands, the total annual cost has been reported as €410.7 million [10].

Although intersectional crashes are generally more frequent than non-intersectional ones, in 2020, 64% of fatal crashes involving cyclists occurred on road segments, defined as areas 20 meters away from intersections, whereas only 26% of such fatalities occurred at intersections [11]. Bil et al. demonstrated that car drivers, when at fault for crashes, often cause more serious consequences for cyclists on straight road sections [12]. In crashes occurring on road segments, several factors contribute to high injury severity, including being in a rural region with an elevated speed limit, male gender, and cyclist age of >55 years [13]. Another identified risk factor is bicycling on roads against oncoming traffic [14].

Although relevant research has shed light on risk factors for bicycle crashes at intersections, few studies have explicitly investigated crashes on road segments. Bicycle crashes on road segments remain a substantial issue for public health concern. This study aims to fill a critical gap by conducting a thorough examination of the risk factors associated with three distinct bicycle crash types: overtaking, rear-end, and door crashes that occur on road segments. Studies that have examined bicycle crashes relatively broadly, without distinguishing crash types, have identified several key factors—including vehicle volume [15], traffic density [16], number of lanes [16], access points along road segments [15], shoulder and median widths [15], parking space availability [15,16], length of continuous two-way left-turn lanes [15], and pavement type [17]—all of which contribute to bicycle crashes on road segments. One notable study has examined the risk factors for overtaking, rear-end, and door crashes [18]. Specifically, Pai identified buses and coaches as common crash partners in overtaking crashes, poor visibility, traversing manoeuvres, and teenage cyclists as risk factors for rear-end crashes, and built-up areas as a risk factor for door crashes [18]. In addition, another study linked the speed of a passing vehicle to increased severity of cyclist injury in overtaking crashes [19]. The high mortality rate from crashes on road segments underscores the significant risks linked to overtaking, rear-end, and door crashes. Overtaking, involving high-speed manoeuvres, greatly increases the likelihood of severe accidents. Rear-end crashes, frequently triggered by sudden stops or aggressive tailgating, pose a persistent threat to cyclists. Furthermore, injuries sustained by cyclists striking an opening car door can be devastating due to the impacts from the door, ground, or vehicles behind. These critical issues highlight the urgent need for identifying risk factors for these crashes.

The primary objective of the present study, an extension of our previous study, was to analyse police-reported crash data from additional years to determine whether the risk factors for these three crash types remained unchanged. The study addresses a critical gap in current research, focusing on crashes specifically occurring on road segments. Existing literature offers limited insights into these crash types, highlighting a crucial need for targeted investigations. These crashes have the potential for severe impacts, involving complex dynamics that demand a nuanced understanding for effective mitigation strategies. By exploring these factors, our research aims to significantly enhance cyclist safety within this particular context. Furthermore, we aimed to untangle the joint associations of several factors—including light conditions, urban versus rural settings, vehicle types, and rider and driver characteristics—with these three crash types.

## Material and methods

### Crash data source

The present investigation utilised data from 01/01/1991 to 31/12/2020, obtained from the United Kingdom's official road traffic casualty database, STATS19. Police record such data either at crash scenes or within 30 days of each crash. The UK's Department for Transport compiles the data, which the United Kingdom Data Archive then maintains and distributes.

The dataset encompasses a variety of variables, including crash circumstances (e.g., time and date, weather conditions, road and light conditions, posted speed limit, road type), vehicle and driver characteristics, demographic details of the drivers, precrash manoeuvres of the vehicles, and the initial impact point of the vehicle. Additionally, the dataset contains demographic information and details regarding injury severity for each casualty. This study adhered to the STROBE (strengthening the reporting of observational studies in epidemiology) reporting guidelines [20]. It was conducted in accordance with the Declaration of Helsinki and received approval from the Joint Institutional Review Board of Taipei Medical University (N202011030).

Injury severity in the aforementioned dataset is divided into three categories, namely slight, serious, and fatal. Fatal injuries refer to those leading to death within 30 days of the accident. Serious injuries include conditions such as fractures, internal injuries, severe cuts and lacerations, concussions, and any injury requiring hospitalisation. Slight injuries include sprains, bruises, and minor cuts, as well as mild shock requiring roadside attention. The exclusive focus of this study was crashes leading to cyclist casualties.

As shown in Fig 1, this study analysed 1,366,196 crashes involving bicycles and other vehicles. Initially, 1,235,032 junction cases were excluded. From the remaining 131,164 bicycle segment crashes, 3,527 were further excluded because of incomplete demographic data for the

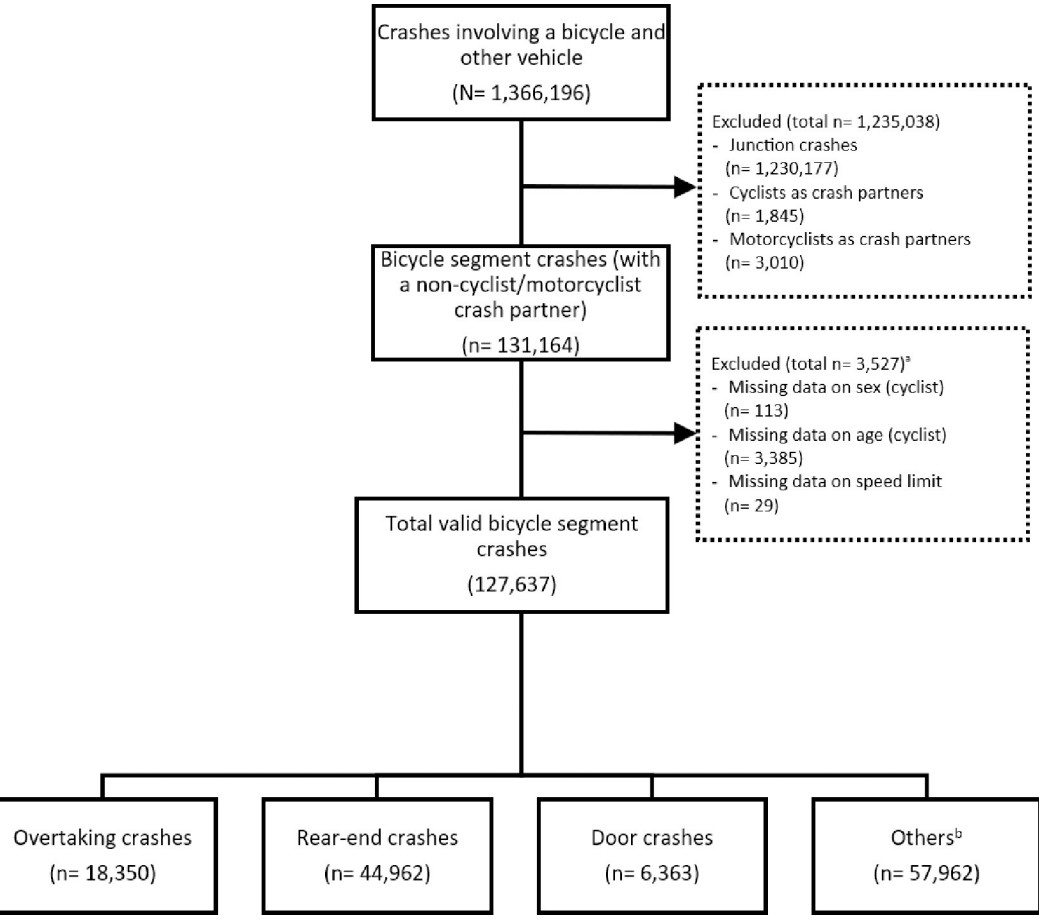

**Fig 1. Flowchart of the study sample selection process.** (a) Listed excluded criteria are nonexclusive; thus, the sum of the total may exceed 3,527. (b) Other crashes include reversing crashes and head-on crashes.

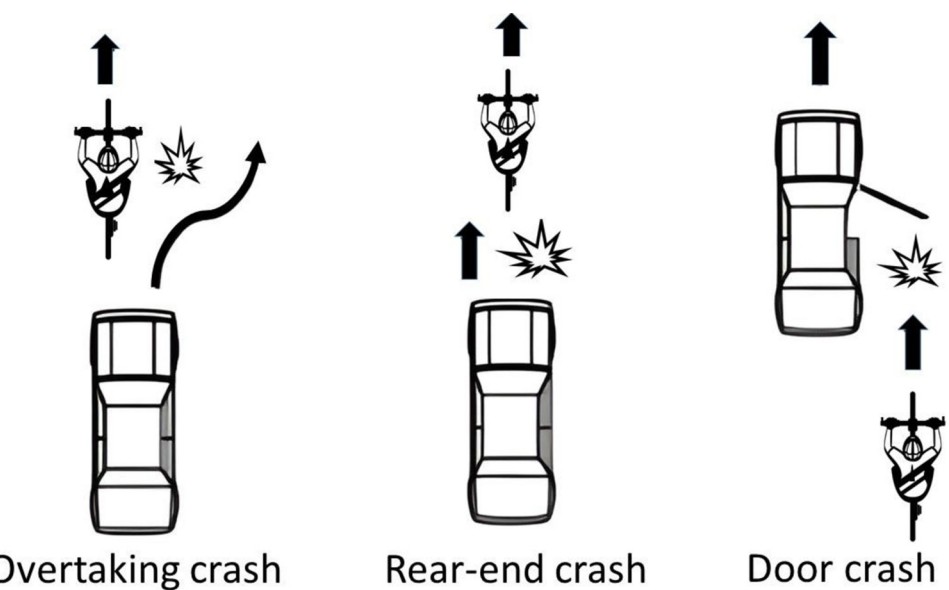

**Fig 2. Illustrative diagram of the three crash types.**

cyclist and missing speed limit information, leaving a valid cohort of 127,637 bicycle segment crashes for analysis. Within this cohort, this study identified 18,350 overtaking crashes, 44,962 rear-end crashes, 6,363 door crashes, and 57,962 other types of crashes.

## Classification of crash types

As shown in Fig 2, an overtaking crash is defined as a crash where a motorised vehicle overtakes and impacts with a bicycle, which may be travelling straight, overtaking another vehicle, changing lanes, or turning. A rear-end crash occurs when a following vehicle impacts with the rear of a bicycle. A door crash involves a bicycle either being struck by or striking the opening door of an automobile. These three crash types were described using schematics in our previous study [18].

## Data analysis

For the present study, the three crash types of focus (overtaking, rear-end, and door crashes) were the binary-dependent variables. The collected data encompassed the following factors: lighting conditions on the roadway at the time of the crash (daylight, darkness-lit, darkness-unlit), the speed limit at the crash scene (rural: $\geq$40 miles per hour [mph]; urban: 20–30 mph), the time of day categorised into four periods according to traffic volume (midnight: 00:00–06:00; rush hours: 07:00–08:00 and 17:00–18:00; nonrush hours: 09:00–16:00; and evening: 19:00–23:00), and the day of the week (weekday or weekend day). The demographic details of cyclist casualties encompassed age ($\leq$18, 19–40, 41–64, or $\geq$65 years) and sex (male or female). Finally, the demographic details of the crash partner included the type of vehicle (identified as a taxi, private hire car, car, bus, or heavy goods vehicle [HGV]), age ($\leq$18, 19–40, 41–64, or $\geq$65 years) and sex (male or female). On a cautionary note, we removed junction cases to avoid the variability introduced when exogenous factors, such as junction geometry and control measures, are present at junctions. Furthermore, the cases involving other cyclists and motorcyclists were removed as we focused on vehicle-cycle crashes only. Missing data on sex, age, or speed limits were also excluded in the analysis. Excluding these data may impact our

results in a marginal scale, as these data are likely to be single-bicycle crashes that in nature be underreported in police crash dataset [21].

## Statistical analysis

This study employed the Chi-squared test to examine the associations between crash type and other factors, including cyclist or motorist characteristics, vehicle features, roadway conditions, and temporal variables. We initially utilized descriptive statistics to examine the distribution of crash types across various variables such as lighting conditions, speed limit, time of day, and day of the week. Demographic details concerning cyclist casualties encompassed age and sex, while information about the crash partner included vehicle type, age, and sex. This preliminary analysis provided a general picture of basic characteristics of the data and identification of potential patterns. For inferential analysis, we applied the Chi-squared test to investigate associations between crash type and various factors, including cyclist and motorist characteristics, vehicle features, roadway conditions, and temporal variables. We then estimated crude odds ratios by estimating univariate logistic regression and adjusted odds ratios by multivariate logistic models, respectively. This approach allowed us to identify significant predictors while controlling for potential confounding variables [22].

The multivariate logistic regression model equation was specified as:

$$log\left(\frac{P(Y=1)}{1-P(Y=1)}\right) = \beta_0 + \beta_1 X_1 + \beta_2 X_2$$

where $P(Y = 1)$ denotes the probability of the outcome, $\beta_0, \beta_1, \beta_2, \ldots, \beta_p$ are the coefficients to be estimated, and $X_1, X_2, \ldots, X_p$ represent the predictor variables.

Before estimating the model, assumptions of logistic regression, such as linearity of the logit, absence of multicollinearity, and independence of observations, were evaluated. An odds ratio (OR) greater than 1 indicated a positive association between the independent variable and the occurrence rate, while an OR less than 1 indicated a negative association. An OR of 1 suggested no association between the variables of interest and the outcomes. Additionally, joint effect analysis was employed to assess the risk associated with the combination of variables across the three types of crashes. All statistical analyses were conducted using SPSS Statistics version 25 for Windows (IBM Corp., Armonk, New York, USA). A $p$ value lower than 0.05 in two-tailed tests was considered statistically significant.

## Results

### Population characteristics

Tables 1–3 present the distributions of overtaking, rear-end, and door crashes, respectively, in relation to multiple independent variables. These data revealed that a significant proportion of bicycle crashes occurred in daylight (82.3%), occurred in urban settings (78.5%), occurred during nonrush hours (48.3%), occurred on weekdays (77.5%), involved cyclists aged under 18 years (40.1%), and involved male cyclists (81.3%). Additionally, most crashes involved cars as crash partners (83.6%), and crash partners were predominately aged 19–40 years (38.5%) and were male (76.4%). Table 1 highlights an overrepresentation in bicycle overtaking crashes for certain variables, namely unlit darkness (19.5%), rural areas (24.8%), midnight hours (17.7%), buses or HGVs as crash partners (24.7%), and elderly crash partners (21.5%) and male crash partners (16.0%). These results were revealed to be statistically significant by the chi-squared test ($p < 0.01$).

**Table 1. Distribution of overtaking crashes according to a set of independent variables.**

| Variable | Total (n = 127,637) | Overtaking crashes (n = 18,350) | Non-overtaking crashes (n = 109,287) | χ2 test p value |
|---|---|---|---|---|
| **Light conditions**, n (%) | | | | <0.001 |
| Daylight | 105,053 (82.3) | 15,283 (14.6) | 89,770 (85.5) | |
| Darkness-lit | 16,543 (13.0) | 1,889 (11,4) | 14,654 (88.6) | |
| Darkness-unlit | 6,041 (4.7) | 1,178 (19.5) | 4,863 (80.5) | |
| **Speed limit**, n (%) | | | | <0.001 |
| Rural (≥ 40 mph) | 27,395 (21.5) | 6,805 (24.8) | 20,590 (75.6) | |
| Urban (20–30 mph) | 100,242 (78.5) | 11,545 (11.5) | 88,697 (88.5) | |
| **Crash time (h)**, n (%) | | | | <0.001 |
| Midnight (00:00–06:00) | 4,810 (3.8) | 852 (17.7) | 3,958 (82.3) | |
| Rush hours (07:00–08:00/17:00–18:00) | 41,619 (32.6) | 5,685 (13.7) | 35,934 (86.3) | |
| Nonrush hours (09:00–16:00) | 61,696 (48.3) | 9,386 (15.2) | 52,310 (84.8) | |
| Evening (19:00–23:00) | 19,512 (15.3) | 2,427 (12.4) | 17,085 (87.6) | |
| **Crash day**, n (%) | | | | 0.094 |
| Weekend | 28,730 (22.5) | 4,218 (14.7) | 24,512 (85.2) | |
| Weekday | 98,907 (77.5) | 14,132 (14.3) | 84,775 (85.7) | |
| **Cyclist's age (years)**, n (%) | | | | <0.001 |
| ≤18 | 51,193 (40.1) | 5,220 (10.2) | 45,973 (89.8) | |
| 19–40 | 45,760 (35.9) | 7,108 (15.5) | 38,652 (84.5) | |
| 41–64 | 26,052 (20.4) | 5,012 (19.2) | 21,040 (80.8) | |
| ≥65 | 4,632 (3.6) | 1,010 (21.8) | 3,622 (78.2) | |
| **Cyclist's sex**, n (%) | | | | <0.001 |
| Male | 103,766 (81.3) | 14,746 (14.2) | 89,020 (85.8) | |
| Female | 23,871 (18.7) | 3,604 (15.1) | 20,267 (84.9) | |
| **Crash partner**, n (%) | | | | <0.001 |
| Taxi/Private hire car | 2,588 (2.0) | 208 (8.0) | 2,380 (92.0) | |
| Car | 106,668 (83.6) | 13,599 (12.8) | 93,069 (87.3) | |
| Bus/Heavy goods vehicle | 18,381 (14.4) | 4,543 (24.7) | 13,838 (75.3) | |
| **Crash partner's age (years)**, n (%) | | | | <0.001 |
| ≤18 | 2,415 (1.9) | 281 (11.6) | 2,134 (88.4) | |
| 19–40 | 49,103 (38.5) | 5,398 (11.0) | 43,705 (89.0) | |
| 41–64 | 35,598 (27.9) | 3,973 (11.2) | 31,625 (88.8) | |
| ≥65 | 40,521 (31.8) | 8,698 (21.5) | 31,823 (78.5) | |
| **Crash partner's sex**, n (%) | | | | <0.001 |
| Male | 97,447 (76.4) | 15,584 (16.0) | 81,863 (84.0) | |
| Female | 30,190 (23.8) | 2,766 (9.2) | 27,424 (90.8) | |

Several variables in Table 2 reveal significant differences between rear-end crashes and non-rear-end crashes. Specifically, a higher proportion of rear-end crashes occurred under darkness-unlit conditions (50.2%) compared to darkness-lit conditions (37.5%). Additionally, rear-end crashes were more prevalent in rural areas with speed limits of ≥ 40 mph (43.0%) compared to urban areas with speed limits of 20–30 mph (33.1%). Crashes involving crash partners aged ≥ 65 accounted for 39.7% of rear-end crashes, which was higher compared to other age groups (age 41–64: 33.0% and ≤18: 36.0%). Furthermore, rear-end crashes were more likely to occur during midnight (47.6%) compared to rush hours (36.3%). Taxis or private hire cars were frequently involved in rear-end crashes (42.4%), as were male crash partners (36.8%). These findings highlight the significant influence of various factors on the likelihood of rear-end crashes. Variables such as darkness-unlit conditions, higher speed limits in rural areas, crash time, and characteristics of the crash partner all emerged as significant determinants. Importantly, these associations were statistically significant, as indicated by the Chi-squared test (p < 0.001).

**Table 2. Distribution of rear-end crashes according to a set of independent variables.**

| Variable | Total (n = 127,637) | Rear-end crashes (n = 44,962) | Non-rear-end crashes (n = 82,675) | $\chi 2$ test p value |
|---|---|---|---|---|
| **Light conditions**, n (%) | | | | <0.001 |
| Daylight | 105,053 (82.3) | 35,726 (34.1) | 69,333 (66.0) | |
| Darkness-lit | 16,543 (13.0) | 6,204 (37.5) | 10,339 (63.5) | |
| Darkness-unlit | 6,041 (4.73) | 3,032 (50.19) | 3,003 (49.71) | |
| **Speed limit**, n (%) | | | | <0.001 |
| Rural ($\geq$ 40 mph) | 27,395 (21.5) | 11,788 (43.0) | 15,607 (57.0) | |
| Urban (20–30 mph) | 100,242 (78.5) | 33,174 (33.1) | 67,068 (66.9) | |
| **Crash time (h)**, n (%) | | | | <0.001 |
| Midnight (00:00–06:00) | 4,810 (3.8) | 2,289 (47.6) | 2,521 (52.4) | |
| Rush hours (07:00–08:00/17:00–18:00) | 41,619 (32.6) | 15,089 (36.3) | 26,530 (63.7) | |
| Nonrush hours (09:00–16:00) | 61,696 (48.3) | 20,723 (33.6) | 40,973 (66.4) | |
| Evening (19:00–23:00) | 19,512 (15.3) | 6,861 (36.2) | 12,651 (64.9) | |
| **Crash day**, n (%) | | | | <0.001 |
| Weekend | 28,730 (22.5) | 9,485 (33.0) | 19,245 (67.0) | |
| Weekday | 98,907 (77.5) | 35,477 (35.9) | 63,430 (64.1) | |
| **Cyclist's age (years)**, n (%) | | | | <0.001 |
| $\leq$18 | 51,193 (40.1) | 13,446 (26.3) | 37,747 (73.7) | |
| 19–40 | 45,760 (35.9) | 19,102 (41.7) | 26,658 (58.3) | |
| 41–64 | 26,052 (20.4) | 10,619 (40.8) | 15,433 (59.2) | |
| $\geq$65 | 4,632 (3.6) | 1,795 (38.8) | 2,837 (61.3) | |
| **Cyclist's sex**, n (%) | | | | <0.001 |
| Male | 103,766 (81.3) | 37,175 (35.8) | 66,591 (64.2) | |
| Female | 23,871 (18.7) | 7,787 (32.6) | 16,084 (67.4) | |
| **Crash partner**, n (%) | | | | <0.001 |
| Taxi/Private hire car | 2,588 (2.0) | 1,096 (42.4) | 1,492 (57.7) | |
| Car | 106,668 (83.6) | 37,202 (34.9) | 71,342 (66.9) | |
| Bus/Heavy goods vehicle | 18,381 (14.4) | 6,664 (36.3) | 9,841 (53.5) | |
| **Crash partner's age (years)**, n (%) | | | | <0.001 |
| $\leq$18 | 2,415 (1.9) | 870 (36.0) | 1,545 (64.0) | |
| 19–40 | 49,103 (38.5) | 16,282 (33.2) | 32,821 (66.8) | |
| 41–64 | 35,598 (27.9) | 11,736 (33.0) | 23,862 (67.0) | |
| $\geq$65 | 40,521 (31.8) | 16,074 (40.0) | 24,447 (60.3) | |
| **Crash partner's sex**, n (%) | | | | <0.001 |
| Male | 97,447 (76.6) | 35,828 (36.8) | 61,619 (63.2) | |
| Female | 30,190 (23.7) | 9,134 (30.3) | 21,056 (69.7) | |

As shown in Table 3, several variables can contribute to door crashes involving bicycles. Door crashes predominantly occurred in urban areas with speed limits of 20–30 mph (6.2%), while a significantly lower proportion occurred in rural areas with speed limits $\geq$ 40 mph (0.5%). These crashes were overrepresented during non-rush hours (5.5%) and rush hours (4.9%) compared to evening (4.3%) and midnight (2.4%). Cyclists were more frequently involved in door crashes on weekdays (5.4%) than weekends (3.7%). As many as 8.2% of all female cyclists were involved in door crashes, which is higher than the involvement rate among males (4.2%). Taxi and private hire cars were overinvolved in door crashes (10.6%) compared to cars (5.2%) and buses/heavy goods vehicles (3.1%). Crash partners aged $\leq$18 years (5.2%) and 19–40 years (5.3%) were disproportionately involved in door crashes compared to older age groups, and female crash partners were overrepresented in door crashes (7.4%) compared to males (4.2%). These results were statistically significant, as indicated by the Chi-squared test ($p < 0.001$). They suggest that various factors—including traffic conditions (rural areas, crash time), cyclist demographics (younger age, female), and characteristics

**Table 3. Distribution of door crashes according to a set of independent variables.**

| Variable | Total (n = 127,637) | Door crashes (n = 6,363) | Non-door crashes (n = 121,274) | χ2 test p value |
|---|---|---|---|---|
| **Light conditions**, n (%) | | | | <0.001 |
| Daylight | 105,053 (82.3) | 5,192 (4.9) | 99,861 (95.1) | |
| Darkness-lit | 16,543 (13.0) | 1,031 (6.2) | 15,512 (93.8) | |
| Darkness-unlit | 6,041 (4.7) | 140 (2.3) | 5,901 (97.7) | |
| **Speed limit**, n (%) | | | | <0.001 |
| Rural (≥ 40 mph) | 27,395 (21.5) | 123 (0.5) | 27,272 (99.6) | |
| Urban (20–30 mph) | 100,242 (78.5) | 6,240 (6.2) | 94,002 (93.8) | |
| **Crash time (h)**, n (%) | | | | <0.001 |
| Midnight (00:00–06:00) | 4,810 (3.8) | 113 (2.4) | 4,697 (97.7) | |
| Rush hours (07:00–08:00/17:00–18:00) | 41,619 (32.6) | 2,056 (4.9) | 39,563 (95.1) | |
| Nonrush hours (09:00–16:00) | 61,696 (48.3) | 3,363 (5.5%) | 58,333 (94.6) | |
| Evening (19:00–23:00) | 19,512 (15.3) | 831 (4.3) | 18,681 (95.7) | |
| **Crash day**, n (%) | | | | <0.001 |
| Weekend | 28,730 (22.5) | 1,072 (3.7) | 27,658 (96.3) | |
| Weekday | 98,907 (77.5) | 5,291 (5.4) | 93,616 (94.7) | |
| **Cyclist's age (years)**, n (%) | | | | <0.001 |
| ≤18 | 51,193 (40.1) | 802 (1.6) | 50,391 (98.4) | |
| 19–40 | 45,760 (35.9) | 3,474 (7.6) | 42,286 (93.4) | |
| 41–64 | 26,052 (20.4) | 1,773 (6.8) | 24,279 (93.2) | |
| ≥65 | 4,632 (3.6) | 314 (6.8) | 4,318 (93.2) | |
| **Cyclist's sex**, n (%) | | | | <0.001 |
| Male | 103,766 (81.3) | 4,404 (4.2) | 99,362 (95.8) | |
| Female | 23,871 (18.7) | 1,959 (8.2) | 21,912 (91.8) | |
| **Crash partner**, n (%) | | | | <0.001 |
| Taxi/Private hire car | 2,588 (2.0) | 273 (10.6) | 2,315 (89.5) | |
| Car | 106,668 (83.6) | 5,514 (5.2) | 101,154 (94.8) | |
| Bus/Heavy goods vehicle | 18,381 (14.4) | 576 (3.1) | 17,805 (96.9) | |
| **Crash partner's age (years)**, n (%) | | | | <0.001 |
| ≤18 | 2,415 (1.9) | 1,62 (5.2) | 2,253 (93.3) | |
| 19–40 | 49,103 (38.5) | 2,585 (5.3) | 46,518 (94.7) | |
| 41–64 | 35,598 (27.9) | 1,887 (5.3) | 33,711 (94.7) | |
| ≥65 | 40,521 (31.8) | 1,729 (4.3) | 38,792 (95.7) | |
| **Crash partner's sex**, n (%) | | | | <0.001 |
| Male | 97,447 (76.6) | 4,123 (4.2) | 93,324 (95.8) | |
| Female | 30,190 (23.7) | 2,240 (7.4) | 27,950 (92.6) | |

of the crash partner (taxi/private hire cars)—significantly contribute to the likelihood of door crashes involving cyclists.

## Risk factors for the three crash types

Table 4 presents the results of the univariate logistic regression models. In terms of overtaking crashes, conditions of darkness with lighting (AOR 0.80, 95% CI: 0.77–0.82, p < 0.001) and darkness without lighting (AOR 0.93, 95% CI: 0.89–0.95, p = 0.001) were linked to a reduced likelihood of crashes when compared to daylight conditions. Urban roads with lower speed limits (20–30 mph) significantly reduced the odds of overtaking crashes compared to rural roads (AOR 0.40, 95% CI: 0.37–0.47, p < 0.001). In terms of cyclist demographics, older cyclists (≥65 years) were at a notably higher risk (AOR 1.84, 95% CI: 1.78–1.97, p < 0.001), and male cyclists were more likely to be involved than female cyclists (AOR 1.14, 95% CI: 1.10–1.17, p < 0.001). Additionally, crashes involving buses or heavy goods vehicles (HGVs) increased the likelihood of overtaking crashes (AOR 1.31, 95% CI: 1.24–1.41, p < 0.001).

**Table 4. Univariate logistic regression results.**

| Variable | Overtaking crashes | | Rear-end crashes | | Door crashes | |
|---|---|---|---|---|---|---|
| | AOR (95% CI) | p value | AOR (95% CI) | p value | AOR (95% CI) | p value |
| **Light condition** | | | | | | |
| Daylight | Ref | | Ref | | Ref | |
| Darkness-lit | 0.80 (0.77, 0.82) | <0.001 | 1.11 (1.08, 1.14) | 0.036 | 1.19 (1.17, 1.26) | <0.001 |
| Darkness-unlit | 0.93 (0.89, 0.95) | 0.001 | 1.50 (1.46, 1.56) | <0.001 | 0.74 (0.72, 1.02) | 0.198 |
| **Speed limit** | | | | | | |
| Rural (≥40 mph) | Ref | | Ref | | Ref | |
| Urban (20–30 mph) | 0.40 (0.37, 0.47) | <0.001 | 0.75 (0.73, 0.79) | <0.001 | 15.3 (14.6, 18.1) | <0.001 |
| **Crash time** | | | | | | |
| Midnight | 1.05 (0.97, 1.10) | 0.157 | 1.34 (1.30, 1.39) | <0.001 | 0.39 (0.35, 0.47) | <0.001 |
| Rush hours | 1.04 (0.98, 1.08) | 0.116 | 1.16 (1.12, 1.20) | 0.003 | 1.36 (1.31, 1.55) | <0.001 |
| Nonrush hours | 1.12 (1.06, 1.14) | 0.007 | 1.02 (0.97, 1.13) | 0.742 | 1.78 (1.68, 1.89) | <0.001 |
| Evening | Ref | | Ref | | Ref | |
| **Crash day** | | | | | | |
| Weekend | Ref | | Ref | | Ref | |
| Weekday | 0.92 (0.90, 1.04) | 0.341 | 1.08 (1.07, 1.13) | <0.001 | 1.33 (1.25, 1.36) | <0.001 |
| **Cyclist's age (years)** | | | | | | |
| ≤18 | Ref | | Ref | | Ref | |
| 19–40 | 1.28 (1.23, 1.39) | <0.001 | 1.80 (1.76, 1.99) | <0.001 | 5.26 (5.20, 5.86) | <0.001 |
| 41–64 | 1.47 (1.33, 1.61) | <0.001 | 1.68 (1.64, 1.81) | <0.001 | 5.66 (5.47, 6.00) | <0.001 |
| ≥65 | 1.84 (1.78, 1.97) | <0.001 | 1.54 (1.51, 1.80) | <0.001 | 5.13 (5.01, 5.83) | <0.001 |
| **Cyclist's sex** | | | | | | |
| Male | Ref | | Ref | | Ref | |
| Female | 1.14 (1.10, 1.17) | <0.001 | 0.81 (0.79, 0.91) | <0.001 | 1.48 (1.33, 1.67) | <0.001 |
| **Crash partner** | | | | | | |
| Taxi/Private hire car | 0.63 (0.641, 0.680) | <0.001 | 1.27 (1.24, 1.334) | <0.001 | 1.78 (1.46, 1.82) | <0.001 |
| Car | Ref | <0.001 | Ref | <0.001 | Ref | <0.001 |
| Bus/HGV | 1.31 (1.24, 1.41) | | 1.05 (1.01, 1.15) | | 0.433 (0.40, 0.51) | |
| **Crash partner's age (years)** | | | | | | |
| ≤18 | 1.03 (0.97, 1.21) | 0.251 | 1.15 (1.11, 1.34) | <0.001 | 0.65 (0.62, 0.69) | <0.001 |
| 19–40 | Ref | 0.035 | Ref | 0.138 | Ref | <0.001 |
| 41–64 | 0.93 (0.91, 0.98) | <0.001 | 0.98 (0.97, 1.03) | <0.001 | 0.96 (0.94, 0.99) | <0.001 |
| ≥65 | 2.33 (1.99, 2.56) | | 1.25 (1.20, 1.31) | | 0.51 (0.47, 0.56) | |
| **Crash partner's sex** | | | | | | |
| Male | 1.28 (1.25, 1.33) | <0.001 | 1.23 (1.15, 1.39) | <0.001 | 1.30 (1.25, 1.53) | <0.001 |
| Female | Ref | | Ref | | Ref | |

For rear-end crashes, both lit (AOR 1.11, 95% CI: 1.08–1.14, p = 0.036) and unlit (AOR 1.50, 95% CI: 1.46–1.56, p < 0.001) darkness conditions were associated with a higher likelihood of crashes compared to daylight. Urban areas were linked to a decreased risk of rear-end crashes compared to rural areas (AOR 0.75, 95% CI: 0.73–0.79, p < 0.001). The likelihood of rear-end crashes was significantly higher during midnight (AOR 1.34, 95% CI: 1.30–1.39, p < 0.001) and rush hours (AOR 1.16, 95% CI: 1.12–1.20, p = 0.003). As with overtaking crashes, older cyclists had an elevated risk (AOR 1.54, 95% CI: 1.51–1.80, p < 0.001), while males had slightly reduced odds compared to females (AOR 0.81, 95% CI: 0.79–0.91, p < 0.001). Crashes involving buses or heavy goods vehicles were slightly more likely to result in rear-end crashes (AOR 1.05, 95% CI: 1.01–1.15, p < 0.001).

Regarding door crashes, lit conditions during darkness were associated with increased odds of crashes (AOR 1.19, 95% CI: 1.17–1.26, p < 0.001), whereas unlit conditions did not show a significant difference compared to daylight (AOR 0.74, 95% CI: 0.72–1.02, p = 0.198). Urban environments with lower speed limits were strongly linked to a higher risk of door crashes

(AOR 15.3, 95% CI: 14.6–18.1, p < 0.001). Older cyclists (≥65 years) faced a substantially increased risk (AOR 5.13, 95% CI: 5.01–5.83, p < 0.001), and male cyclists were more likely to be involved than females (AOR 1.48, 95% CI: 1.33–1.67, p < 0.001). Interestingly, crashes involving buses or heavy goods vehicles reduced the likelihood of door crashes compared to cars (AOR 0.433, 95% CI: 0.40–0.51, p < 0.001).

Table 5 presents the results of the multivariate logistic regression analysis. In overtaking crashes, the presence of HGVs as partners increases the likelihood by 1.3 times (AOR = 1.30, 95% CI = 1.27–1.33; p < 0.001). For cyclists aged 65 and older, the adjusted odds ratio (AOR) is 1.79 (95% CI = 1.65–1.93; p < 0.001) compared to those aged 18 and younger. Factors associated with a decreased likelihood of crashes include daylight conditions (AOR = 0.81, 95% CI = 0.80–0.84; p < 0.001) and rural areas with speed limits of 40 mph or higher (AOR = 0.45, 95% CI = 0.43–0.47; p < 0.001).

For rear-end crashes, significant risk factors included darkness and unlit conditions (AOR = 1.49, 95% confidence interval [CI] = 1.40–1.57; p < 0.001), crashes occurring on

**Table 5. Multivariate logistic regression results.**

| Variable | Overtaking crashes | | Rear-end crashes | | Door crashes | |
|---|---|---|---|---|---|---|
| | AOR (95% CI) | *p* value | AOR (95% CI) | *p* value | AOR (95% CI) | *p* value |
| **Light condition** | | | | | | |
| Daylight | Ref | | Ref | | Ref | |
| Darkness-lit | 0.81 (0.80, 0.84) | <0.001 | 1.04 (1.00, 1.09) | 0.041 | 1.23 (1.20, 1.24) | <0.001 |
| Darkness-unlit | 0.92 (0.90, 0.93) | 0.001 | 1.49 (1.40, 1.57) | <0.001 | 0.87 (0.86, 1.02) | 0.136 |
| **Speed limit** | | | | | | |
| Rural (≥40 mph) | Ref | | Ref | | Ref | |
| Urban (20–30 mph) | 0.45 (0.43, 0.47) | <0.001 | 0.76 (0.74, 0.79) | <0.001 | 16.2 (13.5, 19.4) | <0.001 |
| **Crash time** | | | | | | |
| Midnight | 1.07 (0.98, 1.17) | 0.119 | 1.28 (1.21, 1.40) | <0.001 | 0.50 (0.46, 0.53) | <0.001 |
| Rush hours | 1.06 (1.00, 1.12) | 0.043 | 1.12 (1.09, 1.15) | <0.001 | 1.49 (1.45, 1.62) | <0.001 |
| Nonrush hours | 1.09 (1.03, 1.15) | 0.003 | 1.01 (0.96, 1.10) | 0.639 | 1.90 (1.81, 1.93) | <0.001 |
| Evening | Ref | | Ref | | Ref | |
| **Crash day** | | | | | | |
| Weekend | Ref | | Ref | | Ref | |
| Weekday | 0.97 (0.96, 1.01) | 0.133 | 1.09 (1.06, 1.12) | <0.001 | 1.25 (1.16, 1.34) | <0.001 |
| **Cyclist's age (years)** | | | | | | |
| ≤18 | Ref | | Ref | | Ref | |
| 19–40 | 1.29 (1.24, 1.35) | <0.001 | 1.84 (1.79, 1.89) | <0.001 | 5.94 (5.49, 6.44) | <0.001 |
| 41–64 | 1.51 (1.44, 1.58) | <0.001 | 1.73 (1.68, 1.79) | <0.001 | 6.13 (5.62, 6.68) | <0.001 |
| ≥65 | 1.79 (1.65, 1.93) | <0.001 | 1.67 (1.57, 1.78) | <0.001 | 5.99 (5.22, 6.87) | <0.001 |
| **Cyclist's sex** | | | | | | |
| Male | Ref | | Ref | | Ref | |
| Female | 1.11 (1.06, 1.15) | <0.001 | 0.85 (0.83, 0.90) | <0.001 | 1.68 (1.58, 1.77) | <0.001 |
| **Crash partner** | | | | | | |
| Taxi/Private hire car | 0.64 (0.61, 0.69) | <0.001 | 1.29 (1.19, 1.39) | <0.001 | 1.61 (1.59, 1.69) | <0.001 |
| Car | Ref | <0.001 | Ref | <0.001 | Ref | <0.001 |
| Bus/HGV | 1.30 (1.27, 1.33) | | 1.10 (1.06, 1.14) | | 0.48 (0.45, 0.49) | |
| **Crash partner's age (years)** | | | | | | |
| ≤18 | 1.10 (0.96, 1.25) | 0.162 | 1.19 (1.17, 1.24) | <0.001 | 0.65 (0.63, 0.68) | <0.001 |
| 19–40 | Ref | 0.025 | Ref | 0.026 | Ref | <0.001 |
| 41–64 | 0.95 (0.91, 0.99) | <0.001 | 0.96 (0.95, 0.98) | <0.001 | 0.95 (0.93, 0.98) | <0.001 |
| ≥65 | 2.01 (1.94, 2.09) | | 1.20 (1.18, 1.31) | | 0.54 (0.52, 0.57) | |
| **Crash partner's sex** | | | | | | |
| Male | 1.35 (1.29, 1.42) | <0.001 | 1.15 (1.12, 1.19) | <0.001 | 1.37 (1.30, 1.46) | <0.001 |
| Female | Ref | | Ref | | Ref | |

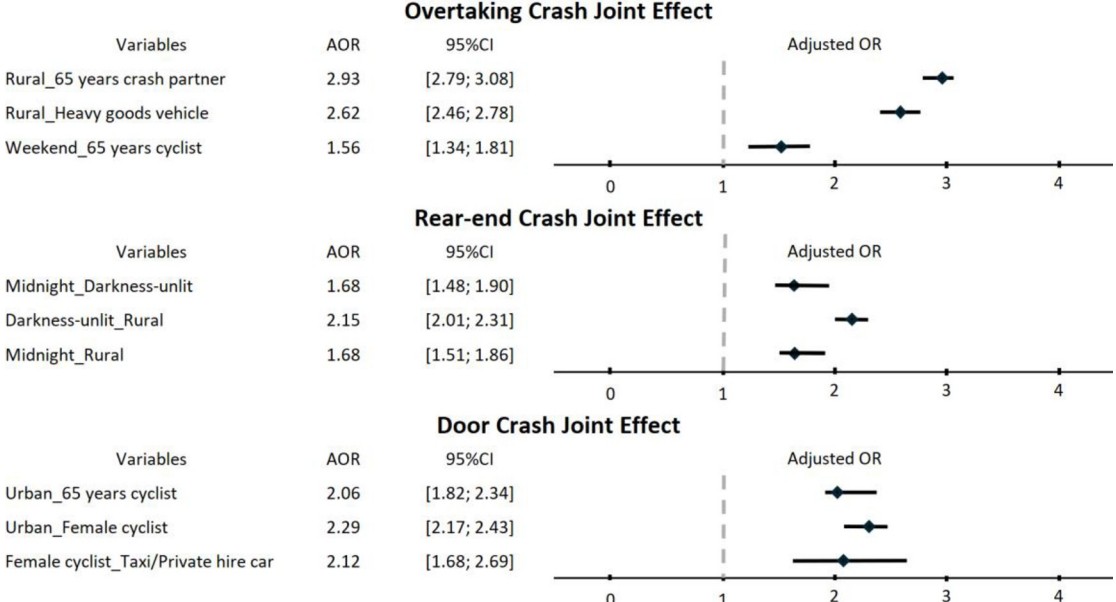

**Fig 3. Joint effects of several variables on the three crash types.**

weekdays (AOR = 1.09, 95% CI = 1.06–1.12; p < 0.001), and an increased likelihood of rear-end crashes during rush hours (AOR = 1.12, 95% CI = 1.09–1.15; p < 0.001). In contrast, the risk is lower in urban areas (AOR = 0.76, 95% CI = 0.74–0.79; p < 0.001) when rural areas are used as the reference.

Door crashes are significantly more prevalent in urban areas with speed limits of 20 to 30 mph—approximately 16 times higher (AOR = 16.2, 95% CI = 13.5–19.4; p < 0.001). Additionally, interactions with taxis or private hire cars as crash partners further increase the likelihood of these crashes (AOR = 1.61, 95% CI = 1.59–1.69; p < 0.001). Other important risk factors include conditions of darkness with illumination (AOR = 1.23, 95% CI = 1.20–1.24; p < 0.001) and crashes occurring on weekdays (AOR = 1.25, 95% CI = 1.16–1.34; p < 0.001). Furthermore, male crash partners were associated with increased odds of door crashes (AOR = 1.37, 95% CI = 1.30–1.47; p < 0.001).

Fig 3 presents a forest plot demonstrating the joint effects of several variables on the three crash types when other variables were controlled for. The results identified several key risk factors for both overtaking and rear-end crashes. The risk of overtaking crashes showed a significant increase of 193% in rural areas when elderly drivers were involved (AOR = 2.93, 95% CI = 2.79–3.08), and similarly when heavy goods vehicles (HGVs) were the crash partner (AOR = 2.62, 95% CI = 2.46–2.78). Elderly cyclists also faced a higher risk of overtaking crashes on weekends (AOR = 1.56, 95% CI = 1.34–1.81).

Regarding rear-end crashes, the risk increased notably with unlit darkness during midnight (AOR = 1.68, 95% CI = 1.48–1.90) and was significantly higher in rural areas (AOR = 2.15, 95% CI = 2.01–2.31). Furthermore, bicycling at midnight in rural areas was associated with an increased risk of rear-end crashes (AOR = 1.68; 95% CI = 1.51–1.86). In urban settings, the risk of door crashes was higher for female cyclists (AOR = 2.29; 95% CI = 2.17–2.43) and for elderly cyclists (AOR = 2.06; 95% CI = 1.82–2.34). Finally, female cyclists exhibited a 112% higher likelihood of door crashes when the crash partner was a taxi (AOR = 2.12; 95% CI = 1.68–2.69).

## Discussion

This study explored the relationships among individual and environmental factors in relation to three common bicycle crash types (overtaking, rear-end, and door crashes) on roads in the United Kingdom from 1991 to 2020. The findings revealed several significant factors. First, for overtaking crashes, HGVs as crash partners, rural areas, and the involvement of elderly crash partners emerged as key contributing factors. Second, unlit darkness, midnight hours, and rural areas were the factors most closely associated with rear-end crashes. Third, urban areas and taxis as crash partners significantly increased the likelihood of door crashes. Moreover, male crash partners were found to be a consistent risk factor across all three crash types.

Our research findings identified specific risk factors for overtaking crashes, namely rural areas, HGVs as crash partners, and elderly crash partners. These findings align with previous research that identified elderly drivers [23], speeds exceeding 10 mph, and the presence of pick-up trucks as factors contributing to increased risk for overtaking crash. Specifically, HGVs possess several characteristics that amplify this danger. Their large blind spots make it difficult for drivers to see cyclists, increasing the likelihood of crashes during overtaking [24]. Additionally, HGVs are less manoeuvrable compared to passenger cars, which reduces their ability to avoid crashes if cyclists suddenly enter their path [25]. The speed and distance perception issues between HGVs and cyclists further complicate the judgment of safe overtaking gaps [26]. Furthermore, HGVs require longer stopping distances due to their size and weight, which can lead to severe consequences if a sudden need to brake arises. A behavioural study suggested that compared with cars, HGVs tended to maintain a narrower clearance zone when overtaking bicycles [27]. Regarding the association with buses or HGVs, Pai et al. suggested that time pressures on HGV drivers for timely loading and unloading might lead to more reckless driving [18]. Specifically, our results align with the observations made by Pai et al., who also mentioned higher crash rates involving buses or HGVs, supporting the idea that these time pressures contribute to increased crash risks. Our findings underscore the necessity of implementing measures such as 'Share the Road' warning signs [28], particularly in rural settings, where HGVs are likely to execute overtaking manoeuvres at high speed. Such measures could prompt motor vehicles to maintain safer distances from the edges of travel lanes, especially in areas with a notable presence of both HGVs and bicycles.

We also identified elderly drivers as a factor contributing to overtaking crashes—a finding consistent with relevant research [23]. We found that as individuals age, their risk of being involved in road accidents increases, primarily due to declines in cognitive capabilities. Our study corroborates these findings by showing that older cyclists are more susceptible to accidents during overtaking manoeuvres, which can be attributed to diminished reaction times and impaired decision-making abilities [29], their health [30], and their driving performance [31]. Notably, crashes involving elderly individuals often occur in scenarios with challenging conditions, including at intersections without traffic control measures, on high-speed roads, during adverse weather conditions, in poorly lit areas, and in head-on accidents [32–34]. The heightened level of risk under such conditions may be attributed to cognitive and perceptual decline in older drivers, which could affect their capacity to execute actions such as overtaking manoeuvres safely. Accordingly, developing specialised cognitive training programmes as interventions to enhance road safety for elderly drivers is evidently necessary [35]. Based on our study's findings, we recommend the development of specialised interventions to improve road safety for elderly cyclists. Our analysis reveals that older cyclists are at a higher risk of being involved in overtaking crashes, with this increased risk being strongly linked to declines in cognitive capabilities associated with aging. To address this issue, we advocate for the implementation of targeted cognitive training programs specifically designed for elderly cyclists.

These programs should focus on enhancing critical skills such as reaction time, situational awareness, and decision-making abilities, which are crucial for reducing crash risk and improving overall road safety.

In the present study, several factors were found to increase the risk of rear-end crashes on road segments, including darkness with unlit surroundings, midnight hours, and rural settings (speed limit > 40 mph). Although few studies have specifically addressed rear-end crashes involving bicycles on road segments, available data suggest that the low conspicuity of bicycles, especially at night, is a recurrent factor in rear-end crashes [18]. Moreover, a lack of adequate street lighting, which is common in rural settings, predisposes cyclists to rear-end crashes. Our joint-effects analysis further indicated that the detrimental effect of unlit darkness is more pronounced in rural areas and during midnight hours. Potential intervention strategies to mitigate rear-end crashes include enhancing illumination and executing speed control management on rural road segments with heavy bicycle traffic.

Next, our analysis successfully identified associations of urban areas and taxis and private hire cars as crash partners with door crashes on road segments. Although research specifically focusing on door crashes on road segments is limited, similar findings were documented by Pai, indicating that urban roadways and taxis contributed to door crashes [18]. However, determining the factors influencing this trend poses a challenge. One possible explanation could be the increased presence of taxis or private hire cars in such areas, where passengers often disembark. Additionally, our analysis further revealed an elevated risk of door crashes involving crashes with taxis in urban areas. To reduce door crashes on road segments, educating taxi drivers, as well as passengers, about the importance of vigilance when opening doors near traffic is essential [18]. In addition, cyclists should be advised to maintain at least a door's width distance from all parked cars to improve the sight triangles of drivers and increase the visibility of cyclists [36]. Implementing a two-stage door opening mechanism for vehicles, which would enable drivers to verify the presence of bicycles to the rear, could also be beneficial [37].

The strengths of this study include the use of STATS19 datasets spanning from 1991 to 2020, which provides a robust statistical foundation and a broad perspective on trends in bicycle crashes. By focusing specifically on three crash types on road segments—overtaking, rear-end, and door crashes—the study provides a comprehensive and focused analysis, which can yield more actionable insights and more effective recommendations. The UK-based dataset ensures that the findings are particularly relevant for local policy and safety interventions. Additionally, the application of statistical techniques and the consideration of various factors, such as crash partner and time of day, enhance the validity and depth of the analysis.

This study had several limitations that warrant acknowledgement. First, the substantial underreporting of nonfatal casualties to the police, particularly casualties involving cyclists not obligated to report accidents, is a critical factor to consider. Such underreporting, as highlighted by the U.K. Government's Department for Transport [11], likely results in the incomplete representation of nonfatal and 'slight' casualties in road casualty data. Second, the STATS19 data utilised in this study lack critical variables, including precrash speeds, specific geometric characteristics of roadways, data regarding alcohol and illicit substance use, and cyclist speed at the time of an accident. Moreover, critical exposure data—such as those related to traffic flow, rider or driver experience, and other elements of risk exposure—are absent, and the absence of such details limits our ability to fully account for potential variations resulting from unobserved factors in the analyses. Finally, this study did not explore annual trends in each type of bicycle crash over the 30-year study period; investigating such trends could provide insights regarding changing behaviours among cyclists and motor vehicle drivers as well as the effects of legislative changes for road speed limits.

One inherent problem with police-reported crash data is the variables not readily available, hereby causing unobserved heterogeneity across the observations. To overcome such a limitation, we estimated separate regression models, as suggested by Kim et al. [38], for the three crash types; such an approach provides greater explanatory power compared to single overall models. Further, we conducted joint-effect analyses of several variables of interest that capture heterogeneity. In our previous studies, we adopted the above-mentioned approaches to overcome the inherent problem with a success [39,40].

Future research directions could involve integrating GPS (Global Positioning System) data and weather conditions to analyse both injury frequency and fatalities of bicycle crashes on road segments. Additionally, exploring the potential of autonomous vehicles for detecting approaching bicycles for door-crashes and implementing AI-controlled lighting systems in rural areas for cyclist detection could be promising areas for further study.

## Recommendations

For overtaking crashes, we recommend implementing 'Share the Road' warning signs, especially in rural areas, and developing specialized cognitive training programs for elderly drivers. Regarding rear-end crashes, our suggestions include improving illumination during night time and implementing speed control measures on rural road segments. For door crashes involving parked cars, we propose enhancing driver sight triangles and increasing cyclist visibility. Moreover, implementing a two-stage door opening mechanism and an automatic detection device in vehicles to alert drivers of bicycles approaching from behind could potentially be beneficial.

## Conclusions

This study identified several significant risk factors for the three predominate types of crashes involving cyclists on road segments: HGVs as crash partners, elderly crash partners, and rural areas for overtaking crashes; unlit darkness, midnight hours, and rural areas for rear-end crashes; and urban areas and taxis as crash partners for door crashes. These risk factors remained unchanged since our previous study conducted in 2011 [18]. The present research enhances the field of bicycle safety research by concluding that the detrimental effects of certain variables become more pronounced under certain conditions. For example, first, cyclists in rural settings exhibited an elevated risk of overtaking crashes involving HGVs. Second, the rear-end crash risk increases in the combined presence of unlit darkness, midnight hours, and rural areas. Finally, in urban settings, the likelihood of door crashes increases when a taxi is the crash partner.

## Acknowledgments

This manuscript was edited by Wallace Academic Editing.

## Author Contributions

**Conceptualization:** Chih-Wei Pai.

**Data curation:** Wafaa Saleh, Bayu Satria Wiratama, Akhmad Fajri Widodo, Hui-An Lin.

**Formal analysis:** Wafaa Saleh, Bayu Satria Wiratama, Akhmad Fajri Widodo, Hui-An Lin.

**Funding acquisition:** Cheng-Wei Chan.

**Methodology:** Chun-Chieh Chao, Chih-Wei Pai.

**Supervision:** Li Wei, Yen-Nung Lin, Shou-Chien Hsu, Chih-Wei Pai.

**Validation:** Hon-Ping Ma, Chenyi Chen, Shih Yu Ko, Chih-Wei Pai.

**Writing – original draft:** Chun-Chieh Chao.

**Writing – review & editing:** Chenyi Chen, Akhmad Fajri Widodo.

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
