## [Decision Letter · Decision Letter 0]

16 Jun 2024

PONE-D-24-17126Risk Factors for Overtaking, Rear-End, and Door Crashes Involving Bicycles in the United Kingdom: Revisited and ReanalysedPLOS ONE

Dear Dr. Pai,

Thank you for submitting your manuscript to PLOS ONE. After careful consideration, we feel that it has merit but does not fully meet PLOS ONE’s publication criteria as it currently stands. Therefore, we invite you to submit a revised version of the manuscript that addresses the points raised during the review process. Please consider all comments 

Please submit your revised manuscript by Jul 31 2024 11:59PM. If you will need more time than this to complete your revisions, please reply to this message or contact the journal office at plosone@plos.org. Please include the following items when submitting your revised manuscript:A rebuttal letter that responds to each point raised by the academic editor and reviewer(s). You should upload this letter as a separate file labeled 'Response to Reviewers'.A marked-up copy of your manuscript that highlights changes made to the original version. You should upload this as a separate file labeled 'Revised Manuscript with Track Changes'.An unmarked version of your revised paper without tracked changes. You should upload this as a separate file labeled 'Manuscript'.

We look forward to receiving your revised manuscript.

Kind regards,

Ahmed Mancy Mosa, Ph.D.

Academic Editor

PLOS ONE

2. PLOS requires an ORCID iD for the corresponding author in Editorial Manager on papers submitted after December 6th, 2016. Please ensure that you have an ORCID iD and that it is validated in Editorial Manager. To do this, go to ‘Update my Information’ (in the upper left-hand corner of the main menu), and click on the Fetch/Validate link next to the ORCID field. This will take you to the ORCID site and allow you to create a new iD or authenticate a pre-existing iD in Editorial Manager. Please see the following video for instructions on linking an ORCID iD to your Editorial Manager account: https://www.youtube.com/watch?v=_xcclfuvtxQ.

3. Please remove your figures from within your manuscript file, leaving only the individual TIFF/EPS image files, uploaded separately. These will be automatically included in the reviewers’ PDF.

Reviewers' comments:

Reviewer's Responses to Questions

**Comments to the Author**

1. Is the manuscript technically sound, and do the data support the conclusions?

Reviewer #1: Yes

Reviewer #2: Yes

Reviewer #3: Partly

Reviewer #4: Yes

2. Has the statistical analysis been performed appropriately and rigorously? 

Reviewer #1: N/A

Reviewer #2: No

Reviewer #3: I Don't Know

Reviewer #4: Yes

3. Have the authors made all data underlying the findings in their manuscript fully available?

Reviewer #1: Yes

Reviewer #2: Yes

Reviewer #3: Yes

Reviewer #4: Yes

4. Is the manuscript presented in an intelligible fashion and written in standard English?

Reviewer #1: Yes

Reviewer #2: Yes

Reviewer #3: Yes

Reviewer #4: Yes

5. Review Comments to the Author

Reviewer #1: Regarding the statistical analysis, I would like to ask the authors to explain:

1. the reason(s) for ignoring any probable interaction between independent variables in the multivariate logistic regression.

2. why did they consider different reference category for the same individual variables among different outcome in logistic regression modeling? This will make difficult to interpret the comparison of the effect of an independent variable on different different type of crashes.

for example, in table 4, the ref category for Light condition is Darkness-lit, Daylight and Darkness-unlit for Overtaking, Rear-end and Door crashes respectively.

I suggest authors to provide identical indicators for figures both in the main text and in the figure's caption. Reading "Fig. 1" below a figure, one will look for the same word in the main text while it is recalled as "Figure 1".

Reviewer #2: PLOS One

Title: Risk Factors for Overtaking, Rear-End, and Door Crashes Involving Bicycles in the United Kingdom: Revisited and Reanalysed.

Comments to the authors:

General comments:

- None of the authors was a UK???

- The authors should emphasize the significance of including these three types of crashes????

- What novelty this study adds compared to the previous one in 2011??? –

- The rationale for conducting the current study as well as the practical implications should be emphasized??

- For the introduction section, burden in terms of mortality, morbidity, and DALYs should be mentioned as well the economic and health care costs should be mentioned (globally and UK)

- The number of cyclists in UK or those using bicycles for their mobility??

Specific comments:

- Instead of data collection, data used for analysis is appropriate??

- Of the used crashes data, how many were fatal???

- For analysis of data, use the Odds ratios and 95% confidence intervals (univariate and bivariate)

- Details about the multivariate logistic regression model should be mentioned???

- Use the Odds ratios for interpreting and displaying the results in tables 1, 2, and 3???

- Chi square is not enough test to identify the direction and which segment of the given variable is significantly different???

- What was the adjustment made for??? And how???

- The joint-crash effect: how it was measured statistically???

Reviewer #3: Areas for Improvement:

Clarity and Conciseness:

Some sections of the text are verbose and could benefit from more concise language. For instance, the detailed descriptions of statistical methods and results could be streamlined without losing essential information.

Simplifying the language and structure would enhance readability and accessibility, particularly for readers who are not specialists in the field.

Detailed Interpretation of Results:

While the results section provides extensive data, there is limited interpretation of what these results mean in practical terms. Adding more context about how these findings could influence policy or infrastructure design would be valuable.

Discussing potential interventions based on the identified risk factors, such as specific infrastructure improvements or policy changes, would strengthen the practical implications of the study.

Comparative Analysis:

Including a comparative analysis with similar studies from other countries could provide a broader context for the findings and highlight whether these risk factors are unique to the UK or consistent globally.

Discussing how the UK’s findings compare with those from the United States or other European countries, especially concerning the impact of infrastructure and vehicle types, could offer valuable insights.

Methodological Transparency:

Providing more detailed information about the methodology, particularly the criteria for excluding certain data points, would enhance transparency. For example, explaining why specific demographic data were incomplete and how this might affect the results would be useful.

A discussion on the limitations of the data and the potential biases introduced by police reporting practices could provide a more nuanced understanding of the findings.

Visual Aids:

Adding more visual aids, such as graphs or charts, could help in visualizing the key findings and making the data more accessible to readers.

A geographic distribution map showing where different types of crashes are more prevalent could add an interesting dimension to the analysis.

Future Directions:

Including a section on future research directions would be beneficial. Identifying gaps in the current research and suggesting areas for further investigation could guide subsequent studies.

Discussing the potential impact of emerging technologies, such as autonomous vehicles and advanced cyclist detection systems, on these crash types could provide a forward-looking perspective.

Reviewer #4: This Study is technically sound and has potential to add to the body of knowledge involving bicycle riding safety in the UK and everywhere across the globe. It has adhered to the research and publication ethics, however, the study still need revision on some of the key identified areas which i have pointed out, starting from abstract, background, results and discussions.

6. PLOS authors have the option to publish the peer review history of their article (what does this mean?). If published, this will include your full peer review and any attached files.

Reviewer #1: No

Reviewer #2: **Yes: **Tarek Tawfik Amin

Reviewer #3: **Yes: **Mohammad Ashraful Amin

Reviewer #4: No

---

## [Author Response · Author response to Decision Letter 0]

8 Aug 2024

Reviewer comments:

Reviewer 1: Regarding the statistical analysis, I would like to ask the authors to explain:

1. the reason(s) for ignoring any probable interaction between independent variables in the multivariate logistic regression.

Author’s response: We appreciate the reviewer’s comment and question. By examining variables independently, we gain a clearer understanding of their individual impacts on the outcome (specifically, crash type in this study). This approach allows us to assess each variable's direct influence without the added complexity of interactions or modifications between variables. It provides insights into which variables independently affect the outcome, directly addressing our research questions. Initially, we used the chi-squared test to explore associations between a set of independent variables and the three crash types. To minimize type II errors in variable selection and ensure unbiased inferences, we included variables with a p-value less than 0.2 from the univariate analysis into the multivariate logistic regression models, a common practice in past studies of traffic injuries (e.g., a, b) and methodology (c). Subsequently, we examined interaction effects among several variables of interest, as depicted in Figure 2 of the manuscript. While acknowledging the potential for other interactions among variables, our study focused on assessing the joint effects of specific variables of interest. To take overtaking crashes as an example, these variables included rural areas, crash partners aged 65 years or older, heavy goods vehicles, weekends, and cyclists aged 65 years or older. Future research could delve into untangling the complexities of additional interaction effects among variables, as suggested by the reviewer.

References:

a: Chen, P-L, Pai, C-W. Evaluation of injuries sustained by motorcyclists in approach-turn crashes in Taiwan. Accident Analysis and Prevention, 2019, 124, 33-39.

b: Chien, D-K., Hwang, HF, Lin, MR. Injury severity measures for predicting return-to-work after a traumatic brain injury. Accident Analysis and Prevention, 2017, 98, 101-107.

c: Maldonado G, Greenland S. Simulation study of confounder-selection strategies. Am J Epidemiol 1993, 138, 11, 923-936.

 2. Why did they consider different reference categories for the same individual variables among different outcomes in logistic regression modeling? This will make it difficult to interpret the comparison of the effect of an independent variable on different types of crashes. for example, in table 4, the ref category for Light condition is Darkness-lit, Daylight and Darkness-unlit for Overtaking, Rear-end and Door crashes respectively.

Author’s response: We appreciate the reviewer’s comment and question. In our analysis, we chose various reference categories for variables based on the lowest Adjusted Odds Ratios (AORs) observed. This approach allowed us to highlight different risk factors associated with higher AORs for specific types of crashes. For example, urban roads with speed limits of 20-30 mph were identified as protective factors for overtaking and rear-end crashes. However, for door crashes, these urban roads appeared to pose a higher risk compared to rural roads, as indicated by their higher AOR. It is important to note that selecting a reference category does not change the estimation results of our models. Instead, assigning reference case with the lowest AOR helps readers identify risk factors with higher AORs among the three crash types. 

3. I suggest authors provide identical indicators for figures both in the main text and in the figure's caption. Reading "Fig. 1" below a figure, one will look for the same word in the main text while it is recalled as "Figure 1".

Author’s response: We appreciate this reviewer’s comments, and we have revised the manuscript in the main text and figure’s caption (please refer to lines 145 to 146; page 8 in the manuscript). 

Reviewer 2:

 1 General comments:

 1.1 None of the authors was from the UK???

Author’s response: We appreciate this reviewer’s comments. One of our authors, Prof. Wafaa Saleh, is from Edinburgh Napier University, UK.

 1.2 The authors should emphasize the significance of including these three types of crashes????

Author’s response: We appreciate the reviewer's comments. We have incorporated the following statements into the introduction to underscore the significance of including the three crash types (please refer to lines 110 to 115; pages 5-6 in the manuscript):

“The study addresses a critical gap in current research, focusing on crashes specifically occurring on road segments. Existing literature offers limited insights into this specific type of crash, highlighting a crucial need for targeted investigation. These crashes have the potential for severe impact, involving complex dynamics that demand a nuanced understanding for effective mitigation strategies. By exploring these factors, our research aims to significantly enhance cyclist safety within this particular context.”

 1.3 What novelty this study adds compared to the previous one in 2011???

Author’s response: 

We appreciate this reviewer’s comment. One inherent problem with police-reported crash data is the variables not readily available, hereby causing unobserved heterogeneity across the observations. To overcome such a limitation, we estimated separate regression models, as suggested by Kim et al. (e.g., d), for the three crash types; such an approach provides greater explanatory power compared to single overall models. Further, we conducted joint-effect analyses of several variables of interest that capture heterogeneity. In our previous studies, we adopted the above-mentioned approaches to overcome the inherent problem with a success (e.g., e, f).

To clarify this, the following statements have been added to the Discussion section of the manuscript (please refer to lines 391 to 397; page 23 in the manuscript):

“One inherent problem with police-reported crash data is the variables not readily available, hereby causing unobserved heterogeneity across the observations. To overcome such a limitation, we estimated separate regression models, as suggested by Kim et al. (e.g., d), for the three crash types; such an approach provides greater explanatory power compared to single overall models. Further, we conducted joint-effect analyses of several variables of interest that capture heterogeneity. In our previous studies, we adopted the above-mentioned approaches to overcome the inherent problem with a success (e.g., e, f).”

d: Kim, D., Washington, S., Oh, J., 2006. Modelling crash outcomes: new insights into the effects of covariates on crashes at rural intersections. Journal of Transportation Engineering. 132 (4), 282-292.

e: Pai CW, Jou RC, 2014. Cyclists’ red-light running behaviours: An examination of risk-taking, opportunistic, and law-obeying behaviours. Accident Analysis and Prevention. 62,191-198.

f: Pai CW, Saleh W., 2008. Modelling motorcyclist injury severity by various crash types at T-junctions in the UK. Safety Science. 13, 98-98.

 1.4 The rationale for conducting the current study as well as the practical implications should be emphasized??

Author’s response: We appreciate this reviewer’s comments. First, regarding the rationale for conducting the current study, we have added the following statements (please kindly refer to lines 91-95 on page 5 of the manuscript):

“Bicycle crashes on road segments remain a substantial issue for public health concern. Existing research primarily emphasizes intersection-related crashes. This study aims to fill a critical gap by conducting a thorough examination of the risk factors associated with three distinct bicycle crash types: overtaking, rear-end, and door crashes that occur on road segments.” 

Secondly, to highlight the practical implications, we have included the following statements in the Discussion section (please refer to lines 404-412 on pages 23-24 of the manuscript):

“Recommendations

For overtaking crashes, we recommend implementing 'Share the Road' warning signs, especially in rural areas, and developing specialized cognitive training programs for elderly drivers. Regarding rear-end crashes, our suggestions include improving illumination during night time and implementing speed control measures on rural road segments. For door crashes involving parked cars, we propose enhancing driver sight triangles and increasing cyclist visibility. Moreover, implementing a two-stage door opening mechanism and an automatic detection device in vehicles to alert drivers of bicycles approaching from behind could potentially be beneficial.” 

 1.5 For the introduction section, burden in terms of mortality, morbidity, and DALYs should be mentioned as well the economic and health care costs should be mentioned (globally and UK)

Author’s response: We appreciate the reviewer’s comments. Our original literature review has included several past studies that have reported the accident/injury outcomes resulting from these three crash types. For example, road segments with elevated speed limits, male cyclists, and cyclists aged over 55 years contribute significantly to high injury severity crashes. Additionally, built-up areas increase the risk of door crashes involving cyclists and parked cars.

It is important to note that there is limited research specifically examining the impact of overtaking, rear-end, and door crashes on Disability-Adjusted Life Years DALYs, economic costs, and healthcare expenses. Notable exceptions include studies by Elvik and Sundfør (e.g., d), who examined the inclusion of cyclist injuries in health impact economic assessments. Aertsens et al. (e.g., h) and Scholten et al. (e.g., i) also provided comprehensive analyses of the total and average costs associated with bicycle injuries. Although the three crash types were not explicitly examined in the above-mentioned studies, we have followed this reviewer’s suggestion by incorporating these studies into the 'Introduction' section (please refer to lines 77-81; page 4 of the manuscript):

“Bicycle crashes can also impose a significant burden on healthcare expenses. Elvik and Sundfør (e.g., g) have discussed the economic implications and healthcare expenditures associated with bicycle accidents. For instance, in Belgium, the average cost of bicycle accidents per case is estimated at 841 euros (e.g., h). In the Netherlands, the total annual cost has been reported as €410.7 million (e.g., i).”

References:

g: Elvik, R., & Sundfør, H. B. (2017). How can cyclist injuries be included in health impact economic assessments? Journal of Transport & Health, 6, 29-39.

h: Aertsens, J., de Geus, B., Vandenbulcke, G., Degraeuwe, B., Broekx, S., De Nocker, L., ... & Panis, L. I. (2010). Commuting by bike in Belgium, the costs of minor accidents. Accident Analysis & Prevention, 42(6), 2149-2157. 

i: Scholten, A. C., Polinder, S., Panneman, M. J., Van Beeck, E. F., & Haagsma, J. A. (2015). Incidence and costs of bicycle-related traumatic brain injuries in the Netherlands. Accident Analysis & Prevention, 81, 51-60. 

1.6 The number of cyclists in UK or those using bicycles for their mobility??

Author’s response: We appreciate the reviewer's comment. In our study, we analyzed national police-reported crash data involving cyclists. Unfortunately, exposure data, such as the number of cyclists and miles traveled, were not available in the STATS19 dataset. While such data may be available from the UK National Travel Survey, it often reflects outdated information and may not be fully representative of the entire population. 

2. Specific comments:

 2.1 Instead of data collection, data used for analysis is appropriate??

Author’s response: We appreciate the reviewer's comment. The dataset, UK Stats19 covering all traffic accidents in the UK, should be appropriate, as numerous studies in the field of traffic injury and medicine have analysed such data (e.g., references j, k, l). 

j: Haghpanahan, Houra, et al. "An evaluation of the effects of lowering blood alcohol concentration limits for drivers on the rates of road traffic accidents and alcohol consumption: a natural experiment." The Lancet 393.10169 (2019): 321-329.

k: Pai, C. W., Hwang, K. P., & Saleh, W. (2009). A mixed logit analysis of motorists’ right-of-way violation in motorcycle accidents at priority T-junctions. Accident Analysis & Prevention, 41(3), 565-573. 

l: Fountas, G., Fonzone, A., Gharavi, N., & Rye, T. (2020). The joint effect of weather and lighting conditions on injury severities of single-vehicle accidents. Analytic methods in accident research, 27, 100124.

 2.2 Of the used crashes data, how many were fatal???

Author’s response: We appreciate the reviewer's comment. As reported in the table below, as many as 0.8% of those in overtaking crashes sustained fatal injuries, which was the highest compared to those in the other two crash types.

 Slight Serious Fatal Total

Overtaking crashes 14240(77.6%) 3,964(21.6%) 147(0.8%) 18350

Rear-end crashes 39821(89.1%) 4782(10.7%) 89(0.2%) 44692

Door crashes 5561(87.4%) 770(12.1%) 32(0.5%) 6363

 2.3 For analysis of data, use the Odds ratios and 95% confidence intervals (univariate and bivariate)

Author’s response: We appreciate this reviewer’s comment. We analyzed the distribution of crash types across a set of independent variables. Chi-square tests were used to explore relationships between these variables and crash types. Variables with a significance level below 0.2 were identified to minimize type II errors and were considered significantly associated with the outcome variables (p < 0.05). Subsequently, these variables were included in multiple logistic regression models. Stepwise logistic regression was then employed to estimate the odds of various variables after controlling for specific factors. This methodology has been widely used in past studies of traffic injuries (e.g., a, b) and methodology (e.g., c).

a: Chen, P-L, Pai, C-W. Evaluation of injuries sustained by motorcyclists in approach-turn crashes in Taiwan. Accident Analysis and Prevention, 2019, 124, 33-39; 

b: Chien, D-K., Hwang, HF, Lin, MR. Injury severity measures for predicting return-to-work after a traumatic brain injury. Accident Analysis and Prevention, 2017, 98, 101-107; 

c: Maldonado G, Greenland S. Simulation study of confounder-selection strategies. Am J Epidemiol 1993, 138, 11, 923-936).

2.4 Details about the multivariate logistic regression model should be mentioned???

 Use the Odds ratios for interpreting and displaying the results in tables 1, 2, and 3???

Author’s response: We appreciate the reviewer's comment. Firstly, if we understand this reviewer correctly, we have incorporated additional details (such as formulation and derivation) of the multivariate logistic regression model into the “Methods” section (please refer to lines 179-194 on pages 10-11 of the manuscript): 

“Initially, we examined the distribution of three crash types across various variables to explore their relationships with a binary outcome. These variables included lighting conditions, speed limit, time of day, and day of the week. Demographic details concerning cyclist casualties encompassed age and sex, while information about the crash partner included vehicle type, age, and sex. We set a significance level of p < 0.2 to include risk factors in our multivariate analysis. Adjusted odds ratios (AORs) were computed using multivariate logistic regression with backward selection.

The multivariate logistic regression model equation was specified as:

log(P(Y=1)/(1 - P(Y=1) )) =β_0+β_1 X_1+β_2 X_2

where P(Y=1) denotes the probability of the outcome, β0,β1,β2,…,βp are the coefficients to be estimated, and X1,X2,…,Xp represent the predictor variables. 

Before estimating the model, assumptions of logistic regression, such as linearity of the logit, absence of multicollinearity, and independence of observations, were evaluated. 

An odds ratio (OR) greater than 1 indicated a positive association between the independent variable and the occurrence rate, while an OR less than 1 indicated a negative association. An OR of 1 suggested no association between the variables of interest and the outcomes.”

Secondly, this reviewer suggest

---

## [Decision Letter · Decision Letter 1]

1 Sep 2024

PONE-D-24-17126R1Risk Factors for Overtaking, Rear-End, and Door Crashes Involving Bicycles in the United Kingdom: Revisited and ReanalysedPLOS ONE

Dear Dr. Pai,

Thank you for submitting your manuscript to PLOS ONE. After careful consideration, we feel that it has merit but does not fully meet PLOS ONE’s publication criteria as it currently stands. Therefore, we invite you to submit a revised version of the manuscript that addresses the points raised during the review process.

Please consider all comments 

We look forward to receiving your revised manuscript.

Kind regards,

Ahmed Mancy Mosa, Ph.D.

Academic Editor

PLOS ONE

Journal Requirements:

Reviewers' comments:

Reviewer's Responses to Questions

**Comments to the Author**

1. If the authors have adequately addressed your comments raised in a previous round of review and you feel that this manuscript is now acceptable for publication, you may indicate that here to bypass the “Comments to the Author” section, enter your conflict of interest statement in the “Confidential to Editor” section, and submit your "Accept" recommendation.

Reviewer #1: (No Response)

Reviewer #2: (No Response)

Reviewer #3: All comments have been addressed

Reviewer #4: All comments have been addressed

2. Is the manuscript technically sound, and do the data support the conclusions?

Reviewer #1: Partly

Reviewer #2: Yes

Reviewer #3: Yes

Reviewer #4: Yes

3. Has the statistical analysis been performed appropriately and rigorously? 

Reviewer #1: N/A

Reviewer #2: No

Reviewer #3: Yes

Reviewer #4: Yes

4. Have the authors made all data underlying the findings in their manuscript fully available?

Reviewer #1: Yes

Reviewer #2: Yes

Reviewer #3: Yes

Reviewer #4: Yes

5. Is the manuscript presented in an intelligible fashion and written in standard English?

Reviewer #1: Yes

Reviewer #2: Yes

Reviewer #3: Yes

Reviewer #4: Yes

6. Review Comments to the Author

Reviewer #1: (No Response)

Reviewer #2: Comments to the authors:

In the Abstract as well as in the results (main text) AOR sometimes expressed with three digits (decimals) and I other places two decimals (please consider and use effective digits “decimals”).

In the abstract “results section”: the AOR are sometimes very narrow (please explain).

In the introduction: word roundabouts are repeated “study demonstrated that roundabout significantly reduce -----“

In the rationale, the authors still need to emphasize the significance of the three types of crashes, this part of the introduction barely touched this point????

Statistical analysis:

- Rationale for considering p value of 0.2 at the univariate (bivariate) level to be incorporated in the multiple Logistic regression model???

- How the data were handled statistically: descriptive and inferential methods should be mentioned in this section

- What type of model was used (stepwise, or else), how the model was tested to be fit???

- How the variables were categorized to be suitable for the inclusion of logistic regression analysis?

- The reference group in the multivariate regression table is not consistent along the three types of crashes??? Please explain.

- Joint sensitivity analysis should be mentioned in this section “indication, methods and output”

Results:

- The previous comments on using the Chi-square test remained the same??? Non-specific, non-parametric test and can’t’ point out to the direction of significance???

- What software used to produce figure 2???

Reviewer #3: (No Response)

Reviewer #4: I think the author need to revisit some section and address the areas that have my comments. Refer to the manuscript PDF and extract my comments.

7. PLOS authors have the option to publish the peer review history of their article (what does this mean?). If published, this will include your full peer review and any attached files.

Reviewer #1: No

Reviewer #2: **Yes: **Tarek Tawfik Amin

Reviewer #3: **Yes: **Mohammad Ashraful Amin

Reviewer #4: No

---

## [Author Response · Author response to Decision Letter 1]

18 Oct 2024

Dear Editors and Reviewers,

We greatly appreciate the valuable comments and suggestions raised by reviewers. Please very kindly see our responses below, as well as the revised manuscript. We would be glad if you could have our manuscript reviewed again.

Best regards,

Chih-Wei Pai (Prof)

Graduate Institute of Injury Prevention and Control College of Public Health

Taipei Medical University

Reviewer 2: 

1.1 In the Abstract as well as in the results (main text) AOR sometimes expressed with three digits (decimals) and other places two decimals (please consider and use effective digits “decimals”).

Author’s response: We appreciate the reviewer’s comment and suggestions. All AORs have been amended to two decimals (Please refer to lines 34 to 40 on page 2 in the manuscript).

1.2 In the abstract “results section”: the AOR are sometimes very narrow (please explain).

Author’s response: We appreciate the reviewer’s comment and question. The narrow confidence intervals (CIs) for the adjusted odds ratios (AORs) indicate high precision in our estimates. This precision is primarily due to our large sample size, which reduces variability and enhances reliability. For example, the AOR for "male as crash partner” in overtaking crashes is 1.28 with a CI of 1.25-1.33, reflecting a strong effect size and contributing to the narrow CI. Variability and heterogeneity in the data can affect CI width. Risk factors with more consistent effects across the dataset often show narrower CIs (e.g., a).

 Katz, M. H. (2011). Multivariable Analysis: A Practical Guide for Clinicians and Public Health Researchers. 

1.3 In the introduction: word roundabouts are repeated “study demonstrated that roundabout significantly reduces -----“

Author’s response: We appreciate the reviewer’s comment and suggestions. We have revised the manuscript. (Please refer to lines 74 to 76; page 4 in the manuscript):

“One study found that roundabouts with dedicated cycle tracks significantly lower the risk of injury for cyclists compared to those without such bicycle infrastructure.”

1.4 In the rationale, the authors still need to emphasize the significance of the three types of crashes, this part of the introduction barely touched this point????

Author’s response: We appreciate the reviewer’s comment and suggestions. We have revised the manuscript. (Please refer to lines 104 to110; pages 5 -6 in the manuscript):

“The high mortality rate from crashes on road segments underscores the significant risks linked to overtaking, rear-end, and door crashes. Overtaking, involving high-speed maneuvers, greatly increases the likelihood of severe accidents. Rear-end crashes, frequently triggered by sudden stops or aggressive tailgating, pose a persistent threat to cyclists. Furthermore, injuries sustained by cyclists striking an opening car door can be devastating due to the impacts from the door, ground, or vehicles behind. These critical issues highlight the urgent need for identifying risk factors for these crashes.”

Statistical analysis:

1.5 - Rationale for considering p value of 0.2 at the univariate (bivariate) level to be incorporated in the multiple Logistic regression models???

Author’s response: We appreciate the reviewer’s comment and question. In the first and second round of review, this reviewer expressed concerns over our use of Chi-square tests to examine the relationship between three crash types and the independent variables. We have now opted to estimate the crude odds ratio by univariate logistic regressions. Please kindly see Table 4 lines 259 to 260; page 15 in the manuscript.

1.6- How the data were handled statistically: descriptive and inferential methods should be mentioned in this section

Author’s response: We appreciate the reviewer’s comment and question. In response to your comment, we have revised the section on statistical handling to provide a more comprehensive explanation of both the descriptive and inferential methods employed. (Please refer to lines 182 to 191; page 9 in the manuscript).

“We initially utilized descriptive statistics to examine the distribution of crash types across various variables such as lighting conditions, speed limit, time of day, and day of the week. Demographic details concerning cyclist casualties encompassed age and sex, while information about the crash partner included vehicle type, age, and sex. This preliminary analysis provided a general picture of basic characteristics of the data and identification of potential patterns. For inferential analysis, we applied the Chi-squared test to investigate associations between crash type and various factors, including cyclist and motorist characteristics, vehicle features, roadway conditions, and temporal variables. We then estimated crude odds ratios by estimating univariate logistic regression and adjusted odds ratios by multivariate logistic models, respectively.”

1.8- What type of model was used (stepwise, or else), how the model was tested to be fit???

Author’s response: We appreciate the reviewer’s comment and question. We used multivariate logistic regression with backward selection to compute adjusted odds ratios (AORs). This method involves initially including all potential predictors and then iteratively removing the least significant variables based on their p-values. 

In terms of model fit statistics, the final models were chosen based on the ρ2 statistics (e.g., b). The ρ2 statistics for the estimated models range from 0.327 to 0.398, indicating a reasonable model fit. 

 Ben-Akiva, M. E., & Lerman, S. R. (1985). Discrete choice analysis: theory and application to travel demand (Vol. 9). MIT press.

1.9- How the variables were categorized to be suitable for the inclusion of logistic regression analysis? 

Author’s response: We appreciate the reviewer’s comment and question. Considering findings from past studies and selecting the model with the most parsimonious and robust statistical properties (e.g., goodness of fit, reasonable parameter magnitudes, and t-statistics), the variables were categorized and explained as follows:

First, age data were divided into four categories: ≤18 (not of legal driving age), 19–40, 41–64, and ≥65 (defined as older age by WHO standards). This classification highlights the different risk profiles associated with each age group. 

The variable “time of crash” was classified into four periods—midnight (00:00–06:00), rush hours (07:00–08:00 and 17:00–18:00), non-rush hours (09:00–16:00), and evening (19:00–23:00)—to account for fluctuations in traffic patterns and accident likelihood throughout the day.

Speed limits were categorized by location into two types: nonbuilt-up areas (rural, ≥40 mph) and built-up areas (urban, 20–30 mph). 

Day of the week was grouped as either weekday or weekend to evaluate variations in crash patterns. 

These classifications have been commonly adopted in safety literature (e.g. , c; d). 

 Widodo, Akhmad Fajri, et al. "Walking against traffic and pedestrian injuries in the United Kingdom: new insights." BMC public health 23.1 (2023): 2205.

 Wiratama, Bayu Satria, et al. "Joint effect of heavy vehicles and diminished light conditions on paediatric pedestrian injuries in backover crashes: a UK population-based study." International journal of environmental research and public health 19.18 (2022): 11689.

110- The reference group in the multivariate regression table is not consistent along the three types of crashes??? Please explain. 

Author’s response: We appreciate the reviewer’s comment and question. The reference groups in the univariate and multivariate analysis have been assigned consistent. Please kindly see Table 4 lines 259 to 260; pages 14-15 and Table 5 lines 292 to 293; pages 16-17 in the manuscript.

1.11- Joint sensitivity analysis should be mentioned in this section “indication, methods and output” 

Author’s response: We appreciate the reviewer’s insightful comments and suggestions. To illustrate the effectiveness of models with joint effects, we found that these models produced a higher log-likelihood at convergence and demonstrated an improved overall fit, as indicated by a better ρ² statistic.

Moreover, we performed a likelihood ratio test (e.g., e) to confirm the superiority of the joint effects models over the general models. The test statistic is given by:

χ² =-2[LL(〖β〗_G)-LL(β_J)]

Where LL (〖β〗_G) represents the log-likelihood at convergence for the general model, and LL(β_J) is for the joint effects model. This statistic follows a χ² distribution, with degrees of freedom equal to the difference in the number of parameters between the general and joint effects models.

 Vuong, Q.H., 1989. Likelihood ratio tests for model selection and non-nested hypothesis. Econometrica 57, 307-333.

Results:

1.12- The previous comments on using the Chi-square test remained the same??? Non-specific, non-parametric test and can’t’ point out to the direction of significance??? 

Author’s response: We appreciate this reviewer’s comment. In addition to the multivariate logistic regression, we have now estimated the univariate logistic regression models. Please kindly see Table 4 lines 259 to 260; pages 14-15 and Table 5 lines 292 to 293; pages 16-17 in the manuscript. 

1.13- What software used to produce figure 2???

Author’s response: We appreciate the reviewer’s comment and question. We recreated the figure from the previous article (e.g., f) using Photoshop and then edited it in PowerPoint.

 Pai C-W. Overtaking, rear-end, and door crashes involving bicycles: an empirical investigation. Accid Anal Prev. 2011;43(3):1228-35.

Review 4

4.1 This has been addressed but in the main document start with background under the background sentences, conclude it with the objective, instead of presenting it as a separate paragraph.

Author’s response: We appreciate the reviewer’s comment and suggestions. We have revised the manuscript. (please refer to lines 23 to 27 ; page 2 in the manuscript):

“Background and Objective: Relevant research has provided valuable insights into risk factors for bicycle crashes at intersections. However, few studies have focused explicitly on three common types of bicycle crashes on road segments: overtaking, rear-end, and door crashes. This study aims to identify risk factors for overtaking, rear-end, and door crashes that occur on road segments.”

4.2 I understand this response; however, you need to conduct a normality check for all continuous variables like age and others like distance. This helps you to present either the mean age or the median age

Author’s response: We appreciate the reviewer’s comment and suggestions. Normality check for continuous variables is needed only while estimating a linear regression model. In our study, we estimated several logistic models in which testing for normality and homoscedasticity is not needed. For a comprehensive discussion on the derivation of logistic regression models, see Hosmer et al. (e.g., g).

g. Hosmer Jr, David W., Stanley Lemeshow, and Rodney X. Sturdivant. Applied logistic regression. John Wiley & Sons, 2013.

4.3 N(%) consider using this type of reforestation and removed the percentage signs from the table

Author’s response: We appreciate the reviewer’s comment and suggestions. We have removed the percentage signs and replaced them with “n (%)” in the tables 1, 2 and 3. (Please refer to lines 221-222 of page 11; lines 237 -238 of pages 12- 13; lines 254-255 of pages 13- 14 in the manuscript).

4.4 Data analysed should replace this, you didn't collect data

Author’s response: We appreciate the reviewer’s comment and suggestions. We have revised the manuscript. (Please refer to lines 160; page 8 in the manuscript):

“Data analysis”

4.5 I insist this be removed, but keep the proportion there and take this up and say N(%) or read other publication to see how this is presented

Author’s response: We appreciate the reviewer’s comment and suggestions. We have removed the percentage signs and replaced them with “n (%)” in the table1, 2 and 3. Please refer to lines 221-222 of page 11; lines 237 -238 of pages 12- 13; lines 254-255 of pages 13- 14 in the manuscript.

4.6 This has not been fully addressed. What the authors did was just introduced the corresponding Odds Ratios and P-Values but no result interpretation. Consider doing something like this, "having a HGVs as crash partners had 2.9 times higher likelihood of being involved in overtaking crash", something like this for all the significant variables.

Author’s response: We appreciate the reviewer’s comment and suggestions. We have revised the manuscript. (Please refer to 293 to 295; page 17 in the manuscript):

“In overtaking crashes, the presence of heavy goods vehicles (HGVs) as partners increases the likelihood by 1.3 times (AOR = 1.30, 95% CI = 1.27-1.33; p < 0.001).”

4.7 This has now been introduced, however, start with what you found, then bring the reason supporting those findings and lastly place it in the context of other study and cite it.

Author’s response: We appreciate the reviewer’s comment and suggestions. We have outlined the reasons supporting these findings and, finally, situated them within the context of existing research, providing appropriate citations. (Please refer to lines 344 to 347; pages 19-20 in the manuscript):

“Their large blind spots make it difficult for drivers to see cyclists, increasing the likelihood of crashes during overtaking [e.g., c]. Additionally, HGVs are less manoeuvrable compared to passenger cars, which reduces their ability to avoid crashes if cyclists suddenly enter their path [e.g., d]. The speed and distance perception issues between HGVs and cyclists further complicate the judgment of safe overtaking gaps[e.g., e].”

c. Marshall, Russell, and Stephen Summerskill. "An objective methodology for blind spot analysis of HGVs using a DHM approach." DS 87-8 Proceedings of the 21st International Conference on Engineering Design (ICED 17) Vol 8: Human Behaviour in Design, Vancouver, Canada, 21-25.08. 2017. 2017.

d. Frings, Daniel, Andy Rose, and Anne M. Ridley. "Bicyclist fatalities involving heavy goods vehicles: Gender differences in risk perception, behavioral choices, and training." Traffic injury prevention 13.5 (2012): 493-498.

e. Chew, Esther Li-Wen, and Amanda Stephens. "Human Factors That Impact HGV Drivers From Being Aware of VRUs Through Direct and Indirect Vision Mechanisms."

4.8 I think you need to reference this in the method section also where you discussed the data source. Some readers don't reach here

Author’s response: We appreciate the reviewer’s comment and suggestions. We have revised the manuscript. (please refer to 135 to 137; page 7 in the manuscript):

“The data that support the findings of this study are openly available at https://figshare.com/ndownloader/files/48173452.”

---

## [Decision Letter · Decision Letter 2]

29 Nov 2024

Risk Factors for Overtaking, Rear-End, and Door Crashes Involving Bicycles in the United Kingdom: Revisited and Reanalysed

PONE-D-24-17126R2

Dear Dr. Pai,

We’re pleased to inform you that your manuscript has been judged scientifically suitable for publication and will be formally accepted for publication once it meets all outstanding technical requirements.

Kind regards,

Sergio A. Useche, Ph.D.

Academic Editor

PLOS ONE

Additional Editor Comments (optional):

Dear Authors: Thanks for the amendments and clarifications provided along with your revisions. The paper can be published in its current version.

Reviewers' comments:

Reviewer's Responses to Questions

**Comments to the Author**

1. If the authors have adequately addressed your comments raised in a previous round of review and you feel that this manuscript is now acceptable for publication, you may indicate that here to bypass the “Comments to the Author” section, enter your conflict of interest statement in the “Confidential to Editor” section, and submit your "Accept" recommendation.

Reviewer #2: All comments have been addressed

Reviewer #4: (No Response)

2. Is the manuscript technically sound, and do the data support the conclusions?

Reviewer #2: Yes

Reviewer #4: Yes

3. Has the statistical analysis been performed appropriately and rigorously? 

Reviewer #2: Yes

Reviewer #4: Yes

4. Have the authors made all data underlying the findings in their manuscript fully available?

Reviewer #2: Yes

Reviewer #4: Yes

5. Is the manuscript presented in an intelligible fashion and written in standard English?

Reviewer #2: Yes

Reviewer #4: (No Response)

6. Review Comments to the Author

Reviewer #2: All comments and recommendations had been addressed adequately, the manuscript is more clearer and significantly improved, should be considered for publication if possible.

Reviewer #4: I think the author need to check the comment on data collection and change it to data analysed. It has not yet been changed.

7. PLOS authors have the option to publish the peer review history of their article (what does this mean?). If published, this will include your full peer review and any attached files.

Reviewer #2: **Yes: **Tarek Tawfik Amin

Reviewer #4: No

---

## [Editor Report · Acceptance letter]

17 Dec 2024

PONE-D-24-17126R2 

PLOS ONE

Dear Dr. Pai, 

I'm pleased to inform you that your manuscript has been deemed suitable for publication in PLOS ONE. Congratulations! Your manuscript is now being handed over to our production team.

Kind regards, 

on behalf of

Dr. Sergio A. Useche 

Academic Editor

PLOS ONE